# Development of Native Essential Oils from Forestry Resources in South Korea

**DOI:** 10.3390/life12121995

**Published:** 2022-11-29

**Authors:** Chanjoo Park, Heesung Woo

**Affiliations:** College of Forest and Environmental Sciences, Kangwon National University, Chuncheon 24341, Republic of Korea

**Keywords:** essential oils, endemic species, forest residues, bioactive compounds

## Abstract

South Korea’s forests occupy approximately 70% of the mainland, therefore, there is considerable potential for waste coming from the forest. Extracting essential oils from underutilised biomass is an economic and sustainable method for the production of high-added-value products. These days, countries’ ownership of domestic natural resources is becoming vital, so there is an urgent need for developing the essential oils from native plants. To increase the value of native essential oils in South Korea, the National Institute of Forest Science (NiFoS) established the <Essential oils bank> to develop the native essential oils as well as develop more research infrastructure by sharing information on native essential oils and enhancing their value. We review the selected essential oils which are listed in the <Essential oil bank> from the literature on major chemical constituents, biological activity, and potential uses of essential oils. Those utilising forest resources for commercial essential oil production need to consider the stable supply of plant material in terms of forest management and conservation. Therefore, *Pinaceae* (*Larix kaempferi*, *Pinus densiflora*, *Pinus koraiensis*) and *Cupressaceae (Chamaecyparis obtusa* and *Chamaecyparis pisifera)* could be potential candidates for commercial essential oil as their waste materials are easily obtained from the plantation after forest management. With their unique fragrance and the bioactive compounds in their oils, potential candidates can be utilised in various industry sectors.

## 1. Introduction

South Korea’s forests occupy approximately 70% of the mainland, therefore, there is considerable potential for waste coming from the forest, such as shrubs and diseased or fire-killed trees, as well as their roots, trunks, or branches removed during forest thinning, or their bark, needles, leaves, and even fruits [1,2]. After thinning management, up to 60% of harvested material remains on-site [3]. Organic biomass wastes generated during the relevant industrial process comprise forest by-products (foliage, branches, tops, stem wood, bark, wasted round wood, stumps, and sawdust) and wood-processing industry by-products (bark, log-off cuts, chipper fines, sawdust, shavings, product-off cuts, sander dusts) [4,5,6]. This organic waste contains significant amounts of cellulose, hemicellulose, lignin, pectin, starch, proteins, fats, fatty acids, fatty alcohols, phenols, terpenes, steroids, resin acids, rosin, waxes, polyphenols, suberin, oils, phytosterols, tannins, flavonoids, and phlobaphenes [4,7].

In South Korea, the annual volume generated by forest management is 115,000 tons, and the total amount of leftovers in the forest is 2,483,000 tons. Further, the area of forest management has increased rapidly: 250,000 ha in 2009, 490,000 ha in 2010, 750,000 ha in 2012, and 1,010,000 ha in 2013 [8]. After forest management, forest waste has been used for biomass resources, yet its market value is low [9]. The organic biomass after forest operation could be transformed and valorised according to the concept of responsible and sustainable forest management [10]. Hence, forestry waste could be a suitable source to obtain products with a high added value, such as bioactive compounds from extractives. There exist good prospects and potential for sustainable and cost-effective management from the residual foliage of forestry to essential oils (EO). For example, *Eucalyptus* species are grown for poles/posts, timber, or pulp, and eucalyptus oil from the leaf biomass represents an added source of profits [11].

EO, also known as volatile oils, are defined as volatile aromatic oily compounds that are concentrated from plant material, including but not limited to flowers, leaves, and twigs [12]. The EOs of herbs and their component products from the secondary metabolism of plants have many applications in folk medicine, food flavouring, and preservation, as well as in the fragrance and pharmaceutical industries [13]. Specifically, the market for EO and suitably formulated cosmetic, pharmaceutical, and medical products are in demand, owing to increased interest in complementary therapies and natural alternatives [14]. Currently, there are 3000 EO that have been identified, yet only around 300 EO are being used as commercial products [15]. Approximately more than 250 species belonging to 164 genera and 69 families of aromatic plants are known to exist across the Korean peninsula [16]. However, the EO industry in South Korea highly relies on imports from abroad. Yet, there is a growing necessity to develop native EO in preparation for the Nagoya Protocol, which emphasises countries’ ownership of domestic natural resources. Calvo-Irabien [17] emphasises that the development of a country’s EO production requires comprehensive knowledge of the diversity of native aromatic plant species and their traditional and current uses and the development of technologies for industrial applications together with potential EO-based products. Further, Ekman, et al. [18] emphasise that the recovery and utilisation of forests and processing by-products are top priorities for targeting the sustainable, green, and recycling- and environment-oriented modern forest industry. This paper reviews the screening of the native EO extracted from forest resources and adds value to the aspect of sustainable industry in South Korea. Further, the biological activities and application of EO are summarised.

## 2. Importance of Developing the Native EO in South Korea

The huge production of EO (>70,000 tonnes per annum) is achieved mainly by major cultivators and producers like the USA, Brazil, India, and China. Likewise, Australia, Malaysia, Indonesia, Thailand, Sri Lanka, South Africa, Africa, Egypt, France, Spain, Italy, Germany, Russia, Nepal, Bangladesh, and Pakistan are important contributors to the worldwide production of EO [19,20]. Van de Braak and Leijten [21] reported that approximately 3000 EO are known, of which about 300 are commercially used as flavours and fragrances. Commonly, EO have various biological activities, such as anti-nociceptive, anticancer, anti-inflammatory, penetration-enhancing, insect-repellent, antiviral, and antioxidant effects [22]. Because of the fragrance, flavouring, odorous, and pharmaceutical properties of EO, they play an important role in various industries, such as supplying raw materials for cosmetics, pharmaceuticals, aromatherapy, food flavourings, preservation, and home care business [23]. For example, Artemisinins are derived from extracts of sweet wormwood (*Artemisia annua*) and are well-established for the effective treatment of malaria [24]. Furthermore, manuka oil (*Leptospermum scoparium*) can be used as a natural herbicide, owing to its main active ingredient, leptospermone [25]

The international market for EO is expanding owing to a growing number of consumers interested in safe and eco-friendly products containing EO [26]. The market size of EO in South Korea is estimated to be 150 billion won, yet more than ninety percent of EO used in South Korea are imported from abroad, so there is an urgent need for developing the EO from native plants in Korea [27]. These days, countries’ ownership of domestic natural resources is becoming vital, owing to the Nagoya Protocol. There are articles related to the Access and Benefit-sharing of Genetic Resources (ABS) in the Nagoya Protocol, which consequently acknowledges each country’s ownership of its domestic natural resources [28]. In particular, to promote the equitable sharing of profits, ABS provides a framework by which researchers seeking genetic, protein, or small molecule resources from biodiverse countries could gain access to these biomaterials while also compensating the country of origin should a subsequent product become profitable [29]. To resolve these issues, it is essential to discover and develop commercial essential oils from native plants in South Korea.

To increase the value of native EO in South Korea, the National Institute of Forest Science (NiFoS) established the <Essential oils bank> to develop the native EO as well as develop more research infrastructure by sharing the information on native EO and enhancing their value [30]. This report includes a total of 50 native essential oil crops in South Korea by providing a comprehensive overview of EO and highlighting their potential for development. According to the <Essential oils bank>, there are a total of 39 aromatic crops from forest resources, and the most common volatile crops in South Korea are *Pinaceae* (Table 1). Lee, et al. [31] explained that EO from conifers were recognised as safe, natural disinfectants for various applications, and their chemical profile and antimicrobial activities have been extensively investigated worldwide. *Pinus* essential oil was found to contain aromatic constituents, such as α-terpineol, α-pinene, and *B*-caryophyllene, which show antimicrobial activities [32,33].

Thus, the development of native EO could raise Korea’s share in the global EO market and boost the EO industry in South Korea.

## 3. Development of Commercial EO from Forestry Resources

Nearly 3000 EO have been produced from at least 2000 plant species, out of which 300 are important from a commercial point of view [34]. To become commercial essential oil crops, systematic cultivation strategies for essential oil bearing plants are necessary to ascertain the steady supply of quality plant materials for essential oil production [35].

Commercial production of EO from forest resources is potentially advantageous because (1) they are an environmentally friendly crop, (2) they have been grown naturally, and (3) they are considered potentially profitable, and the development of a sustainable crop may contribute to the protection of native wild germplasm, which can provide valuable material for crop improvement [36]. The commercialisation of medicinal plants might cause issues for the survival of some of the plants [11]. Owing to wild harvests, production could put pressure on the available stocks, thereby threatening the species through overharvesting [37].

Those utilising forest resources for commercial EO production need to consider the stable supply of plant material in terms of forest management and conservation. In South Korea, there could be two major possibilities of unstable plant materials, such as legally protected species and pine wilt disease. First, it is important to prevent rare species from disappearing or extinction instead of developing commercial production such as the indiscriminate harvesting of rare and endemic plant resources. In 2020, the Ministry of Environment (ME) in South Korea designated 377 species of vascular plants on the regional Red List, reflecting the International Union for Conservation of Nature’s (IUCN) Red List for “Guidelines for listing and delisting rare & endangered species and management of endangered Species System” [38]. The IUCN List divides species into nine categories: Not evaluated, Data Deficient (DD), Least Concern (LC), Near Threatened (NT), Vulnerable (VU), Endangered (EN), Critically Endangered (CE), Extinct in the Wild (EW), and Extinct (EX).

As shown in Table 2, these woody plants are not only valuable forestry resources but also aromatic crops. Yet, plants are only distributed in restricted regions in South Korea. Further, alpine and subalpine zones are vulnerable to climate change [39,40]. The long lifespan of trees or narrow-range species shows the difficulties of rapid adaptation to environmental changes [41].

*A. Koreana*, also known as Korean fir, is a conifer species endemic to South Korea [43]. EO extracted from *A. koreana* generate high economic and medical values, owing to their bactericidal and anti-inflammatory effects [44]. Further, the oils show an anti-wrinkle and whitening effect, which can be potentially useful in the development of cosmetic ingredients [45]. However, *A. koreana* have suffered declines in their habitats and suffered from dieback because of increased global warming and artificial destruction since the 1980s [46,47]. For this reason, Korean fir has been assessed as Vulnerable by the International Union of Conservation of Nature (IUCN) and as Endangered by the National Institute of Biological Resources, South Korea. *T. koraiensis* is one precious economic coniferous species with fragrance, ornamental, and medicinal properties, yet it is a critically endangered species in South Korea. Facing global climate change, the natural habitats of *T. koraiensis* have declined owing to competition with temperate species [48]. Fu, et al. [49] found that the oil from the branches and leaves of *T. koraiensis* show an antibacterial effect. *A. nephrolepis* is a fir species growing in high, mountainous areas with small disjunct distributions in South Korea [50]. *J. chinensis* is an ornamental valuable plant and has been used in traditional medicine for the treatment of gout, rheumatism, diarrhoea, and chronic tracheitis [51,52]. However, *J. chinensis* populations are distributed in restricted regions in South Korea, yet this rare plant experiences overcutting and the destruction of its habitats [53,54].

Second, pine forests have become threatened by pine wilt disease (PWD), which causes a significant economic loss. *Bursaphelenchus xylophilus* L. is a highly pathogenic plant parasite that infects mainly *Pinus* species and causes PWD [55]. Likewise, PWD has caused extensive damage in pine forests, in particular in South Korea [56]. Currently, *P. thunbergii* forests in the lowlands of Jeju Island have been continuously damaged by mountain area development, and the damage caused by pine wilt disease is also a major issue together with rapid climate change [57].

Therefore, it is essential to provide a continuous, reliable source of plant material for commercial essential oil production without endangering the natural resource base. Further, studies of breeding and cultivation technologies are essential to secure promising resources for commercial EO.

## 4. Screening the Potential Candidates of EO from Forestry Resources

The “Forest Law” and “Law of Enforcement of Illegal Forest Products” were implemented in 1961 to achieve successful forest rehabilitation. From 1946 to 2000, forest planting projects achieved nearly 83% forest coverage of the total forestland. Artificial forestation especially had a positive influence on the state of current forests. Further, the South Korea government strictly prohibited illegal forest harvesting and illegal shifting cultivation. [58]. Hence, it is fundamental to follow the forest management plan together with the Forest Law to utilise the forest resources in South Korea.

To develop commercial EO, better knowledge and further studies on the production, chemical constituents, and application of EO are needed. Many EO were investigated for their various pharmacological potentials [59]. EO from medicinal as well as other edible plants have been recognised as safe food flavouring agents and aromatic disinfectants with antimicrobial and antioxidant properties [31]. Specifically, EO with bactericidal and fungicidal properties are widely used as alternatives to synthetic chemical products to protect the ecological equilibrium [60]. In addition, EO can be used in the protection of crops from damage by pests and plagues, with the advantage of not accumulating in the environment and having a broad range of activities, which decreases the risk of developing resistant pathogenic strains [61]. Therefore, by utilising the forest biomass, EO could bring new opportunities for the cost-effective and sustainable management of unused forestry biomass [62].

As shown in Table 3, we review the native aromatic crops from forestry resources as potential candidates for commercial EO. The selected EO are listed in the <Essential oils bank> established by NiFoS [30], and four crops are omitted, owing to the unstable supply of plant material (Table 2). The production and chemical profile of EO are dependent on various factors including but not limited to genetic variation, plant nutrition, climate, and agronomic management [26]. Hence, this review covers the literature regarding plant material from South Korea, as chemical constituents of EO are highly affected by genetic and environmental factors, even within the same species [63]. Therefore, major chemical constituents, important pharmacological actions, and the application of potential commercial EO are summarised in Table 3.

### 4.1. EO from Pinaceae

The leaves of the *Pinus* are widely used in folk medicine and food additives, owing to their various pharmacological properties. *Pinus* EO have fresh, herbal, woody, and piney notes. Moreover, EO from conifers are recognised as a safe natural disinfectant for various applications, and the bioactive compounds in the oils together with antimicrobial activities have been extensively investigated [31].

Considering the sufficient forestry resources in South Korea, *Larix kaempferi* (Lamb.) Carrière, *Pinus densiflora* Siebold & Zucc., and *Pinus koraiensis* Siebold & Zucc., could be potential candidates for commercial essential oil. These species are prevalent in South Korea’s landscape, and large amounts of their waste products are obtained from forest management. These three crops are considered recommendable afforestation tree types in South Korea, so most research has been focused on wood, plantation, and forest management [73,74,75]. However, there is limited research on the extracts and EO from these species.

EO from *L. kaempferi* could be used as a natural herbicide, owing to the high percentage of monoterpenes. The major components of EO from *L. kaempferi* are α-pinene (19.86%), β-pinene (17.35%), and L-bornyl acetate (15.29%) [9]. EO from aromatic crops have been reported to contain allelopathic properties, suppressing the growth of other plants [76]. Ben-Ami, et al. [77] found that monoterpene compounds are responsible for germination inhibition. Specifically, monoterpenes in essential oils show phototoxic effects, and they might cause anatomical and physiological changes in plant seedlings, leading to the accumulation of lipid globules in the cytoplasm and reduction in some organelles [78,79,80]. There are various studies on the utilisation of EO for weed management in terms of organic agriculture practices [81,82,83]. Batish et al. [84] found that Eucalyptus oil treatment (0.0714%, *v*/*v*) caused a phytotoxic effect on a noxious weed by showing a rapid electrolyte leakage in the leaf tissues.

The major constituents of EO obtained from the *P. densiflora* wood are longifolene (19.71%), α-terpineol (19.18%), and sabinene (13.53%) [64]. There is no research on EO from the pine needle, yet its extracts have been used for various folk medicinal remedies for treating ailments such as neuralgia, diabetes, hypertension, and skin disease [85]. The chemical constituent longifolene present in *P. densiflora* pine wood oil has shown strong anti-inflammatory activity, owing to the inhibition of degranulation and the expression of cytokines [64,67]. Therefore, *P. densiflora* pine wood oil could be helpful to relieve or prevent allergic diseases.

The essential oil from the leaves of *P. koraiensis* contains α-pinene (21.3%), α-terpineol (11.0%), and δ-3-carene (10.2%) [65]. Joo, et al. [86] found that the essential oil from leaves of *P. koraiensis* has a hypoglycemic potential by inhibiting reactive oxygen species (ROS) and endothelial NO synthase (eNOS) in STZ-treated HIIT-T15 cells as a potent anti-diabetic agent. Hypoglycemic herbal medicines are considered safe for long-term diabetes control owing to their little toxicity [87]. There is limited research on essential oils, yet leaf extract could be used as a cosmetic ingredient, owing to its whitening and anti-wrinkle effects [88,89]. Furthermore, major components from cones’ essential oils are limonene (27.90%), α-pinene (23.89%), and β-pinene (12.02%). Due to the higher amounts of α-pinene, the oils show strong antifungal activity [32].

### 4.2. EO from Cupressaceae

The *Cupressaceae* family are perennial shrubs used for the treatment of neuralgia and are a diuretic and aphrodisiac in folk medicine in Korea, China, and Japan [90]. The plantations of *C. obtusa* and *C. pisifera* have increased since the 1970s, owing to the Park and Landscape management project under the South Korean government. However, the organic biomass after forest operation has not been utilised properly [91].

The main components of *C*. *pisifera* are 3-carene (35.0 %), (−)-bornyl acetate (19.8%), α-pinene (13.0%), myrcene (9.2%), terpinolene (4.9%), and limonene (3.9%) [66]. There is no research on the biological activity of essential oil from *C*. *pisifera* grown in South Korea. Considering the abundant component of 3-carene (35%), this essential oil might show strong fungicidal activity. *Anacyclus valentinus* essential oil contains strong fungicidal activity, owing to the predominance of 3-carene (31%) in this oil [92]. Therefore, *C*. *pisifera* essential oil could be a potential candidate to be used as a safe biocontrol agent to prevent food and crops from fungal disease and improve product quality.

*C. obtusa* is an evergreen coniferous tree that has been planted in the southern province of South Korea, owing to its economic value [93]. In South Korea, there are various products such as sprays, soaps, chopping boards, and cosmetic products that utilize *C. obtusa* extract because of its biological activity together with the relaxing fragrance. Major chemical components of these essential oils are α-cadinol (19.25%), τ-muurolol (14.20%), and α-pinene (13.74%) in *C*. *obtusa* [67]. Among the components of the EO, α-pinene might be active in inhibiting the growth of microorganisms [31]. Ahn et al. [94] found that *C. obtusa* essential oil could be used as a skin protective agent because of its antibacterial, anti-oxidant, and anti-inflammatory effects.

### 4.3. EO from Rutaceae

The *Rutaceae* family contains a rich source of monoterpenes [95]. *Z. schinifolium* (Sancho) and *Z. piperitum* (Chopi) have been commonly used as a favourite spice and as an aromatic medicinal ingredient in South Korea [96,97]. The major compounds in *Z. schinifolium* were found to be estragole (75.03%), 4-methoxybenzaldehyde (4.60%), and 2-undecanone (2.86%) [68]. There is no research on the biological activity of *Z. schinifolium* oil, yet *Z. schinifolium* extract shows antimicrobial activity, suggesting its utilisation as a decontamination agent against *Vibrio parahaemolyticus*, which is a food-borne disease organism [98].

*Z. coreanum* is a Korean lime tree that only grows in the southernmost area of Korea, including Jeju Island. *Z. coreanum* essential oil, which is extracted from fruit, shows reduced melanin production in cells by inhibiting the tyrosinase activity, owing to β-ocimene, α-pinene, and sabinene in the oils [69,99]. *Z. coreanum* essential oil can be safely applied to the skin and may be useful as a potential skin-whitening ingredient in cosmeceuticals. Further study is needed to evaluate the safety of formulations containing the EO of *Z. coreanum* as whitening cosmeceuticals for preventing side effects.

### 4.4. EO from Magnoliaceae

*M. kobus* is a medicinal plant with a wide distribution throughout Korea, and the flower buds of this plant have been used as a folk medicine for the treatment of headaches, nasal obstruction in colds, and toothaches [100]. The major components in its oils are 3-carene (77.07%), *B*-elemene (6.92%), and caryophyllene (2.86%). *M. kobus* oil has potential as a cosmetic functional material containing antimicrobial and anti-inflammatory effects. [70].

### 4.5. EO from Verbenas

*V. rotundifolia* grows wild in the southern part of Korea and along the coast of the West Sea. Since ancient times, it has been used as a folk medicine or herbal medicine [101]. The major components in the oils, which are extracted from leaves, are 1,8-cineole (19.89%), α-terpineol (7.94%), and manoyl oxide (2.40%) [71]. The EO can be useful in cosmetic applications such as aromatherapy, as natural products possessing anti-inflammatory efficacy [72].

## 5. The Two Potential Candidate Essential Oil Families for Healthy Products

*Pinaceae* (*L. kaempferi*, *P. densiflora*, *P. koraiensis*) and *Cupressaceae (C. obtusa* and *C. pisifera*) could be potential candidates for commercial essential oils. These species are prevalent in South Korea’s landscape, and large amounts of their waste products are obtained from forest management. Barbieri and Borsotto [102] explained that the demand comes from the following markets: food and beverage (35%), fragrances, cosmetics and aromatherapy (29%), household (16%), and pharmaceutical (15%). *Pinus* EO have fresh, herbal, woody, and piney notes. Moreover, EO from conifers are recognised as a safe, natural disinfectant for various applications, and the bioactive compounds in their oils together with antimicrobial and anti-inflammatory activities have been extensively investigated [9,31,64]. In particular, *P. koraiensis* oil could be used as an ingredient in functional food for the management of diabetes, as it shows reduced blood glucose and low-density lipoprotein oxidation in streptozotocin-induced diabetic mice [65]. Further research is needed to elucidate and quantify the various biological activities of the selected EO for utilising EO products, including the verification of various biological activities and safety efficacy. 

## 6. Limitations and Prospects

The EO extracted from seeds, flowers, stems, and roots generally contain 0.1–10% fresh weight (*v*/*w*) EO and often <0.1%. Sometimes, up to 20% *v*/*w* or even higher amounts of EO in plant tissues have also been found [103]. Hence, it is important to choose a high-yield EO crop and maximise the oil yield for commercial production. Nonetheless, there is little data available in the literature to indicate the oil yield of potential native EO from forestry resources (Table 3). Furthermore, such data are needed to compare the oil yield among the candidates and optimise the yield and composition of the selected commercial EO.

EO are regarded as GRAS (generally regarded as safe) grade chemicals by the U.S. Food and Drug Administration (FDA); as such, there are no toxicity issues associated with them [104]. In general, the use of EO means less toxicity, reduced genotoxicity (even after long-term use), an ability to act on multiple cellular targets, and a low cost of production [105]. In cosmetics and perfumes, the amount of added EO should not exceed 5%, whereas in food, it should be <0.1% [106]. Some EO and their components have been known to cause allergic contact dermatitis in people who use them frequently [107,108]. Most of the studies have focused on the identification of chemical constituents in EO together with their biological activities; yet, there are limited studies on the mechanism of the biological effects of EO. Therefore, it is imperative to carry out these studies. Further, it is needed to study the safety efficacy of EO. Further, EO are recommended to be used very carefully with considerable precautions about the concentrations being used, product application, the consumer, and major constituents of the oil and toxicology profile [109].

In South Korea, studies on plant extracts are blooming rapidly in the food and cosmetic industry, owing to plant extracts’ biological activity and safety efficacy. Cho, et al. [110] emphasised that natural substances are safer for the human body than antibiotics and chemicals and are even susceptible to resistant bacteria. Yet, there is limited research on EO compared to plant extracts. Further, native aromatic plants in Korea generally simply utilise indigenous specialties from a country or a region. Therefore, there is a need for actively developing commercial EO together with the utilisation of EO. This paper reviews the screening of native EO extracted from forest resources and adds value in the aspect of sustainable industry. At this point, our study contributes to the better valorisation of native EO from forestry resources in South Korea.

## 7. Summary and Conclusions

This review attempts to highlight the potential of EO utilising forest resources as new bioactive constituents and their therapeutic potential on the basis of current scientific studies. In particular, converting unattended forest waste resources into value-added products, such as EO, could bring new opportunities for the cost-effective and sustainable management of unused forestry biomass. In South Korea, there are several plant species that contribute to the significant amount of residual biomass resulting from forest management and contain substantial quantities of EO with a high value. Specifically, *Pinaceae* (*L. kaempferi*, *P. densiflora*, *P. koraiensis*) and *Cupressaceae* (*C. obtusa* and *C. pisifera*) could be potential candidates for commercial essential oils. These species are prevalent in the South Korean landscape, and large amounts of their waste products are obtained from forest management. Notably, the potential candidate EO demonstrate various biological activities, so the selected EO show great potential for the development of pharmaceutical, cosmetic, food, and agrochemical products. In light of this vital importance, this study aimed to perform a review of the literature with a particular emphasis on developing native EO, taking into consideration the cost-effective and sustainable management of unused forestry biomass.

## Figures and Tables

**Table 1 life-12-01995-t001:** Native aromatic crops in South Korea [30].

Family		Scientific Names	Common Names	Part Used
*Pinaceae*	1	*Pinus thunbergii* Parl.	Black pine	leaves
	2	*Abies koreana* E.H. Wilson	Korean fir	leaves
	3	*Larix kaempferi* (Lamb.) Carriere	Japanese larch	branches and leaves
	4	*Picea abies* (L.) H. Karst.	Norway spruce	branches and leaves
	5	*Pinus rigida* Mill	Pitch pine	leaves
	6	*Pinus densiflora* for. multicaulis	Many-stem Korean red pine	leaves
	7	*Abies nephrolepis* (Trautv. ex Maxim.) Maxim.	Khingan fir	leaves
	8	*Pinus parviflora* Siebold & Zucc.	Ulleungdo white pine	branches
	9	*Machilus japonica* Siebold & Zucc.	Long-leaf bay-tree	leaves
	10	*Pinus densiflora* Siebold & Zucc.	Korean red pine	leaves
	11	*Tsuga sieboldii* Carriere	Ulleungdo hemlock	leaves
	12	*Pinus strobus* L.	White pine	leaves
	13	*Pinus koraiensis* Siebold & Zucc.	Korean pine	leaves
	14	*Abies holophylla* Maxim.	Needle fir	leaves
	15	*Picea koraiensis* Nakai	Korean spruce	leaves
*Cupressaceae*	16	*Juniperus rigida* Siebold & Zucc.	Needle juniper	leaves
	17	*Thuja koraiensis* Nakai	Korean arborvitae	leaves
	18	*Juniperus chinensis* var. sargentii A. Henry	Dwarf juniper	leaves
	19	*Chamaecyparis pisifera* cv. Filifera Aurea	Oriental arborvitae	leaves
	20	*Thuja orientalis*	Korean arborvitae	leaves
	21	*Chamaecyparis obtusa* (Siebold & Zucc.) Endl.	Hinoki cypress	branches and leaves
	22	*Juniperus chinensis* L.	Chinese juniper	leaves
	23	*Chamaecyparis pisifera* (Siebold & Zucc.) Endl.	Chamaecyparis pisifera	leaves
*Lauraceae*	24	*Cinnamomum camphora* (L.) J. Presl	Camphor tree	leaves
	25	*Neolitsea sericea* (Blume) Koidz.	Irregular-streak newlitsea	leaves
	26	*Lindera obtusiloba* Blume	Blunt-lobe spicebush	leaves
	27	*Cinnamomum yabunikkei* H. Ohba	Japanese camphor tree	leaves
	28	*Cinnamomum loureirii* nees	Cinnamomum cassia	leaves
	29	*Neolitsea sericea* (Blume) Koidz.	Sericeous newlitsea	leaves
*Rutaceae*	30	*Zanthoxylum ailanthoides* Siebold & Zucc.	Alianthus-like prickly-ash	fruit
	31	*Citrus reticulata* Blanco.	pseudogulgul	fruit
	32	*Zanthoxylum schinifolium* Siebold & Zucc.	Mastic-leaf prickly ash	fruit
	33	*Orixa japonica* Thunb.	East Asian orixa	leaves
	34	*Citrus unshiu* (Yu. Tanaka ex Swingle) Marcow.	Unishiu orange	fruit
	35	*Zanthoxylum coreanum* Nakai	Large-leaflet prickly-ash	fruit
*Cryptomeria*	36	*Cryptomeria japonica* (Thunb. ex L.f.) D. Don	Japanese cedar	leaves
*Lamiaceae*	37	*Agastache rugosa* (Fisch. & CA Mey.) Kuntze	Korean mint	twigs and leaves
*Magnoliaceae*	38	*Magnolia kobus* DC.	Kobus manolia	flower
*Verbenas*	39	*Vitex rotundifolia* L.f.	Beach vitex	fruit

**Table 2 life-12-01995-t002:** Native aromatic crops in South Korea.

Scientific Names	Red Data Book of Republic of Korea ^(1)^	IUCN Red List ^(2)^
*Abies koreana* E.H.Wilson	Endangered (EN)	Vulnerable (VU)
*Thuja koraiensis* Nakai	Vulnerable (VU)	Vulnerable (VU)
*Abies nephrolepis* (Trautv. ex Maxim.) Maxim.	Endangered (EN)	Least Concern (LC)
*Juniperus chinensis* L.	Vulnerable (VU)	Least Concern (LC)

Red Data Book of Republic of Korea ^(1)^: Red Data Book of Republic of Korea (volume 5, Vascular plants) was published by the National Institute of Biological Resources [42]. IUCN Red list ^(2)^: International Union for Conservation of Nature’s (IUCN) Red List for “Guidelines for listing and delisting rare & endangered species and management of endangered Species System” [38].

**Table 3 life-12-01995-t003:** Chemical profile and application of potential commercial native EO from forestry resources, according to the species and organs used for extraction.

Family	Scientific Names	Plant Parts	Major Chemical Constituents	Biological Activity	Potential Use	References
*Pinaceae*	*Larix kaempferi* (Lamb.) Carriere	leaves	α-pinene (19.86%), β-pinene (17.35%), L-bornyl acetate (15.29%)	herbicidal activity	agriculture industry for an herbicidal purpose	Yun, Cho, Yeon, Choi, and Kim [9]
		wood	α-pinene (18.57%), α-cadinol (6.24%), cembrene (6.12%)	anti-inflammatory effect	interior renovations which can effectively improve allergic inflammation	Yang et al. [64]
	*Pinus densiflora* Siebold & Zucc.	wood	longifolene (19.71%), α-terpineol (19.18%), sabinene (13.53%)	anti-inflammatory activity	pharmaceutical industry for relieving the allergy	Yang, Choi, Jeung, Kim, and Park [64]
	*Pinus koraiensis* Siebold & Zucc	leaves	α-pinene (21.3%), α-terpineol (11.0%), δ-3-carene (10.2%)	anti-hyperlipidemia and antidiabetic effects	pharmaceutical industry as an ingredient in functional food	Kim et al. [65]
		cones	limonene (27.90%), α-pinene(23.89%), β-pinene (12.02%),	antimicrobial activity	environmentally friendly disinfectant	Lee, Yang, Lee, and Hong [32]
*Cupressaceae*	*Chamaecyparis pisifera* var. filifera	leaves	3-carene (35.0 %), (−)-bornyl acetate (19.8 %), α-pinene (13.0 %)	-	-	Kim and Lee [66]
	*Chamaecyparis obtusa*	leaves	α-cadinol (19.25%), τ-muurolol (14.20%), α-pinene (13.74%)	anti-inflammatory	pharmaceutical industry for relieving allergy	Yang et al. [67]
*Rutaceae*	*Zanthoxylum schinifolium* Siebold &Zucc.	fruit	estragole (75.03%), 4-methoxybenzaldehyde (4.60%), 2-undecanone (2.86%)	insecticidal activity	agriculture and food industry for controlling mites	Lee [68]
	*Zanthoxylum coreanum* Nakai	fruit	β-ocimene (24.48%), α-pinene (16.56%), sabinene (10.81%)	whitening	cosmetic functional material	Kim et al. [69]
*Magnoliaceae*	*Magnolia kobus*	flower	3-carene (77.07%), β-elemene (6.92%), caryophyllene (2.86%)	antibacterial and anti-inflammatory activity	cosmetic functional material	Lee et al. [70]
*Verbenas*	*Vitex rotundifolia* L. fil.	leaves	1,8-cineole (19.89%), α-terpineol (7.94%), manoyl oxide (2.40%)	anti-inflammatory activity	cosmetic industry as an ingredient for anti-inflammatory efficacy	Jang et al. [71,72]

## Data Availability

Not applicable.

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
