# Peer review of "Development of Native Essential Oils from Forestry Resources in South Korea"

_life, 2022, doi:10.3390/life12121995_

Round 1

Reviewer 1 Report

The article has been well discussed and is quite complete. Suggestions to complete this article are as follows: Add subtitles about materials and methods, a more specific explanation regarding the benefits of the results of this research and discussion for scientific and industrial development, add further research that needs to be done in the future.

Author Response

Response to Reviewer 1 Comments

: Thank you very much for your consideration, and we really appreciate the comments that have resulted in an improved manuscript. Changes have been made and are highlighted (blue) in the revised manuscript according to the suggestions of reviewers.

Point 1:  Add subtitles about materials and methods

Response 1: This is the review manuscript. So, there is no materials and methods. We organise the subtitle in our manuscript following the logic.

  1. Introduction
  2. Importance of developing the native essential oils in South Korea
  3. Development of commercial essential oils from forestry resources
  4. Screening the potential candidates of essential oils from the forestry resources
  5. The two potential candidates essential oil family for the healthy products
  6. Limitation and prospect
  7. Summary and conclusion

Point 2: A more specific explanation regarding the benefits of the results of this research and discussion for scientific and industrial development, add further research that needs to be done in the future.

Response 2: We mentioned the benefits of the results of this research and discussion for scientific and industrial development in <6. Limitation and prospect> (row 364-377). According to your suggestion, we re-structure and revise the content in chapter 6.

  1. Benefits of the research : We add the specific explanation according to the suggestion.
  2. discussion for scientific and industrial development: we have a chapter of industrial development (3. Development of commercial essential oils from forestry resources). So, we cover the content of scientific and industrial development of essential oils and also add the content.
  3. further research that needs to be done in the future: We already mentioned further studies in the manuscript (oil yield research_row 353-355 /safety efficacy of essential oils_row 366 /Mechanism of the biological effect of essetnial oils_row385/)

* We attach the revised manuscript, reflecting the three reviewers' comments.

Reviewer 2 Report

The topic of the review is interesting. However, there are few comments which need to be addressed. 

1. The author repeatedly mentions waste management and forest management.  However, the manuscript only describes sources of essential oils from forest, information which is widely available in various review articles. It would be better if a section on waste resources  is added, i.e. how waste resources is produced and how essential oil can be extracted. 

2. A significant plagiarism is detected. 

3. Please check the referencing style. It should be as per journal specification. 

4. Please address the comments below. 

Author Response

: Thank you very much for your consideration, and we really appreciate the comments that have resulted in an improved manuscript. Changes have been made and are highlighted (red) in the revised manuscript according to the suggestions of reviewers. We checked the reference style.

  1. line 61 Please edit the sentence. ….are known to exist across Korean …..

Response 1: we revised the menioned section (line 61).

  1. Mention the abbreviation of essential oil (EO) at the first instance.

Response 2: we use the abbreviation of essentail oils (EO)_line 49. After then, we use the abbreviation of essential oils (EO) in whole manuscript.

  1. line 85, it plays an important role in various industries

Response 3: we revised the mentioned sentence (line 85)

  1. line 164 However, A koreana suffered decline in their population, because of increased …..

Response 4: we revised the mentioned sentence (line 161)

  1. line 193-194 From 1946 to 2000, forest planting projects achieved nearly 83% forest coverage of total forestland.

Response5: we revised the mentioned sentence (line 190-191)

  1. line 247 change Eucalypt to Eucalyptus

Response 6: we revised the mentioned sentence (line 243)

  1. line 251, essential oils from the pine needle, yet its extract have been …..

Response 7: we revised the mentioned sentence (line 248)

  1. line 252-253, The chemical constituent longifolene present in P. densiflora pinewood oil have shown strong anti-inflammatory activity, owing to inhibition of degranulation and expression of cytokines.

Response 8: we revised the mentioned sentence (line 249-252)

  1. line 256, to relieve or prevent allergic diseases.

Response 9: we revised the mentioned sentence (line 252)

  1. line 261, safe for long-term diabetes …….

Response 10: we revised the mentioned sentence (line 257)

  1. line 263, replace would with could

Response 11: we revised the mentioned sentence (line 259)

  1. line 265, Due to the higher …………

Response 12: we revised the mentioned sentence (line 261)

  1. line 269, used for the treatment ………

Response 13: we revised the mentioned sentence (line 265)

  1. line 280, replace would with could

Response 14: we revised the mentioned sentence (line 276)

  1. line 283-285, In South Korea, there 283 are various products such as sprays, soaps, chopping boards and cosmetic products which utilize C. obtusa extract because  of  its  biological  activity  together  with  the  relaxing  fragrance.

Response 15: we revised the mentioned sentence (line 279-282)

  1. line 302-303, Z. coreanum essential oil which is extracted from the fruits show reduced melanin production in cells through inhibition of tyrosinase activity, owing to the presence of β-ocimene, α-pinene, and sabinene in oils.

Response 16: we revised the mentioned sentence (line 300-302)

  1. line 325, kindly consider revising the heading

Response 17: we agonise the alternative hading. Yet, we decide to keep the heading.

  1. line 321-322, The essential oils can be useful in cosmetic applications such as aromatherapy as natural products possess anti-inflammatory efficacy.

Response 18: we revised the mentioned sentence (line 320-321)

  1. line 335-338, In particular, P. koraiensis oil could be used as an ingredient of functional food for the management of diabetes, as it shows reduced blood glucose and low-density lipoprotein oxidation in streptozotocin-induced diabetic mice.

Response 19: we revised the mentioned sentence (line 334-336).

  1. line 339-341, Further research is  needed  to  elucidate  and  quantify  the  various  biological  activities of  selected essential oils to enhance the safety and efficacy. 

Response 20: we revised the mentioned sentence (line 339-340).

  1. Line 347, Nonetheless, there is little data available in the literature to indicate the oil yield of potential native essential oils from forestry resources

Response 21: we revised the mentioned sentence (line 345-346).

  1. Line 356-358, Most of the studies are focused on the  identification  of  chemical  constituents  in  essential  oils  together  with their biological activities, yet there are limited studies on the safety and efficacy of essential oils. Therefore, it is imperative to carry out these studies.

Response 22: we revised the mentioned sentence (line 355-358).

  1. Line 365-368, Further research should be performed on the safety of essential oils for the human body, changes in the quality of crops treated with  the  oils,  effects  on  natural  enemies  and  formulations  for  improving  367 insecticidal potency and stability

Treated with oil?? Please elaborate. Also clearly mention which natural enemies the author is referring to?

Response 23: we checked the content, and we use the wrong referenece. So, we delete the mentioned sentence. Especially, we re-structure the content in chapter 6.

  1. Line 369, studies on plant extract are booming rapidly in food and cosmetic ……….

Response 24: we revised the mentioned sentence (line 366-367).

  1. Line 372, chemicals and are even susceptible to resistant bacteria

Response 25: we revised the mentioned sentence (line 369).

  1. Line 393, taking into consideration cost-effective and sustainable ……….

Response 26: we revised the mentioned sentence (line 390-391).

* We attach the revised manuscript, reflecting the three reviewers' comments.

Reviewer 3 Report

The relevance of the study of essential oils is clear. The authors in the review have widely shown the possibility of using essential oil resources of forests, focusing on the possible use of waste in the harvesting of essential oil trees and on resource conservation. The authors have made a wide review of the literature on the presented topic. All references in the text correspond to the presented material.

However, some comments have arisen:

1.       Line 134-135 «For example, the demand for the bark from Prunus africanna in Cameroon, almost threatened the remaining natural populations to extinction, owing to the anti-cancer effect from the bark» – it is necessary to confirm this information with a link.

2.       The accent in the manuscript is on essential oils, their resources and application. Therefore, it is unnecessary to provide information that does not correspond to the purpose of the review as in Line 175 («A. nephrolepis is an economically essential species for the production of pulp wood and timber [51]») or other places in the text.

3.       Line 252-253 «Especially, P. densiflora pine wood oil showed strong anti-inflammatory activity, owing to longifolene [65,87]» – check link 87, it describes the antifungal activity of longifolene.

4.       In Table 1 (Native aromatic crops in South Korea) 39 potential sources of essential oils are given. But in Table 3 and further, only 9 types of Chemical profile and application are considered. There is no review of the remaining 30 species and no conclusion whether there is potential for their use in terms of the source of essential oils.

Author Response

Thank you very much for your consideration, and we really appreciate the comments that have resulted in an improved manuscript. Changes have been made and are highlighted (red) in the revised manuscript according to the suggestions of reviewers.

* We attach the revised manuscript, reflecting the three reviewers' comment.

Point 1: Line 134-135 «For example, the demand for the bark from Prunus africanna in Cameroon, almost threatened the remaining natural populations to extinction, owing to the anti-cancer effect from the bark» – it is necessary to confirm this information with a link.

Response 1: We delete the mentioned sentence (row 131-133.)

Point 2: The accent in the manuscript is on essential oils, their resources and application. Therefore, it is unnecessary to provide information that does not correspond to the purpose of the review as in Line 175 («A. nephrolepis is an economically essential species for the production of pulp wood and timber [51]») or other places in the text

Response 2: We delete the sentence which you mentioned (row 171 ).

Point 3:  Line 252-253 «Especially, P. densiflora pine wood oil showed strong anti-inflammatory activity, owing to longifolene [65,87]» – check link 87, it describes the antifungal activity of longifolene.

Response 3: The research of anti-inflammatory activity in P. densiflora pine wood oil is Reference 65. The antifungal activity of P. densiflora pine wood oil is out of topic, so we delete the unnecessary reference. Also, reviewer 2 polish the mentioned section (row 249-251).

 Point 4   In Table 1 (Native aromatic crops in South Korea) 39 potential sources of essential oils are given. But in Table 3 and further, only 9 types of Chemical profile and application are considered. There is no review of the remaining 30 species and no conclusion whether there is potential for their use in terms of the source of essential oils

Response 4: We already mentioned explanation in chpater 4 with table 3. We reviewed the total 39 potential essential oils species from <Essential oils bank>, which is established by the National Institute of Forest Science (South Korea). However, there are limited studies for the overall mentioned aromatic crops in Table 1. Especially, , this review covers the literature regarding plant material from South Korea as chemical constituents of essential oils are highly affected by genetic and environmental factors, even in the same species. Also, we omit the legally protected spices, and vulnerable species from pine wilt disease in South Korea. Because it is essential to provide a continuous, reliable source of plant material for commercial essential oil production. We already mentioned this content in chapter 3. Development of commercial essential oils from forestry resources. Hence, we narrow down the list (Table 1) and we comprehensively provide the review for the potential commercial essential oils crop (Table 3).

Round 2

Reviewer 2 Report

The comments have been addressed. However, the plagiarism still remains. Please refer to attached plag report and act accordingly.
